# Activity of Mono-, Bi-, and Trimetallic Catalysts Pt-Ni-Cr/C in the Bicyclohexyl Dehydrogenation Reaction

**DOI:** 10.3390/molecules27238416

**Published:** 2022-12-01

**Authors:** Alexander N. Kalenchuk, Leonid M. Kustov

**Affiliations:** 1Chemistry Department, Moscow State University, 1 Leninskie Gory, Bldg. 3, 119991 Moscow, Russia; 2N.D. Zelinsky Institute of Organic Chemistry RAS, 47 Leninsky Prosp., 119991 Moscow, Russia; 3Institute of Ecology and Engineering, National University of Science and Technology MISiS, 4 Leninsky Prosp., 119049 Moscow, Russia

**Keywords:** hydrogen storage, bicyclohexyl, dehydrogenation, PtNiCr/C catalysts

## Abstract

The influence of metals with different redox properties and a carbon carrier on the activity of mono-, bi- and trimetallic Pt-Ni-Cr/C catalysts has been studied in the bicyclohexyl dehydrogenation reaction as the hydrogen release stage in hydrogen storage. An increase in the conversion (X > 62%) of bicyclohexyl and selectivity for biphenyl (S > 84%) was observed on trimetallic catalysts Pt-Ni-Cr/C compared with the monometallic catalyst Rt/C (X > 55%; S > 68%). It has been established that the increase in the conversion of bicyclohexyl and selectivity for biphenyl in the dehydrogenation reaction on trimetallic catalysts is due to an increase in the activity of Pt nanoparticles in the vicinity of local Cr-Ni clusters of solid substitution solutions.

## 1. Introduction

Polycyclic naphthenes with high gravimetric hydrogen content (>7 wt.%) have recently attracted increasing attention as potential liquid organic hydrogen carriers for fuel cells [1,2,3,4,5]. The accumulation, storage, transportation and release of hydrogen is carried out by hydrogenation–dehydrogenation of the corresponding aromatic hydrocarbons [6,7,8]. The chemical purity of the stored and released hydrogen without a carbon footprint should be provided by highly active catalysts with high selectivity for the final product, capable of conducting reactions without by-products of hydrogenolysis with ring opening and cracking with the formation of light hydrocarbons. In hydrogenation processes, this condition is met by Pt-, Pd-, and Ni-containing catalysts [9,10,11]. Aluminum or silicon oxides are most often used as carriers, but modern carbon materials combining high strength characteristics and low cracking activity are of greater interest for hydrogen storage purposes [12,13]. At the same time, a simple and effective way to improve the interaction of the active metal with the carbon carrier is the oxidation of the latter. Using the example of carbon nanotubes, it is shown [14,15] that this provides a high concentration of oxygen and a large number of surface defects that contribute to the stabilization of metal particles at the surface. During the oxidation of the Sibunit carbon material in a Pt/C catalyst [16], an increase in dispersion and a decrease in the average size of metal particles were observed, which, during the dehydrogenation of bicyclohexyl, jointly contributed to an increase in the conversion compared with the untreated carrier. Table 1 shows typical examples of mono-, bi, and tricyclic naphthenes, during the dehydrogenation of which high hydrogen release rates were obtained on Pt catalysts, which make it possible to qualitatively compare the activity of substrates with an increase in the number of hydrocarbon cycles in these molecules, even without taking into account the influence of the carrier and the reaction conditions.

The data in Table 1 clearly indicate a significant decrease in the efficiency of hydrogen extraction from naphthenic compounds with an increase in the number of cyclohexane cycles in the molecules. At the same time, a high rate of hydrogen release is achieved at a sufficiently high concentration of Pt, which encourages the search for alternative catalysts with a low content of noble metal without deactivation and loss of activity.

It is known that along with the electronic interaction of metals with the carrier, the higher activity, and selectivity of the supported catalysts are provided by promoting additives [18,19,20,21,22,23,24,25]. In particular, an increase in the adsorption of reacting molecules on the catalyst surface was observed [26,27,28], when platinum was combined with transition metals. The effect seems to be associated with a decrease in the electron density on the 5d orbitals of Pt during the interaction of the electron shells of the two metals. An increase in the number of active metals complicates the electronic processes between them and contributes to the growth of possible topochemical reactions occurring on the surface of catalysts. Nevertheless, some authors [29,30] showed an increase in the platinum activity in a three-component Pt-Ni-Cr catalytic system compared to binary systems, presumably due to an increase in the surface area due to the formation of complex multiphase chromium alloys with Ni crystallites.

The purpose of this work is to study the effect of surface effects during the formation of mono-, bi-, and trimetallic Pt-Ni-Cr catalytic systems deposited on the carbon carrier Sibunit (C) in the reaction of bicyclohexyl dehydrogenation as the hydrogen release stage in hydrogen storage. The study of catalysts was carried out using methods of CO chemisorption, XRD, and TEM, including high-resolution TEM, XPS, TPR, and magnetometry.

## 2. Results and Discussion

To determine the specific surface area and the degree of functionalization of the Sibunit carbon carrier before and after oxidation, the corresponding samples were examined by CO chemisorption and X-ray photoelectron spectroscopy (XPS). The analysis of the obtained data showed (Appendix A, Table 2) that during oxidation, the textural characteristics of the Sibunit deteriorated—the specific surface area (S_BET_) and the total pore volume decreased. At the same time, the average pore size (g_pore_) has increased, which is important for reducing diffusion restrictions during the dehydrogenation of large molecules.

At the same time, with the overall dominance of sp^2^-hybridized carbon atoms (284 eV), about 5% of the total number of carbon atoms are sp^3^-hybridized (289 eV), which indicates that some C-C bonds break at the surface of Sibunit as a result of oxidation. The latter contributes to the expansion of the number and increase in the content of terminal oxygen-containing groups, which, when bicyclohexyl is dehydrogenated on monometallic Pt/C catalysts, leads to an increase in the conversion and selectivity to biphenyl, compared with the untreated carrier (Table 3).

According to CO chemisorption data, the average particle sizes and dispersions of platinum in catalysts 0.1Pt/C and 0.5Pt/C_ox_ are close (d_av(Pt)_~1.7–1.8 nm, D~60%). Micrographs of the 0.1Pt/C_ox_ catalyst (see Appendix A) indicate a uniform distribution of metal over the surface, but HR-TEM images show the presence of single particles with dimensions of about 10 nm at the surface of both catalysts (see Appendix A). However, their HR-TEM photos demonstrate that the diffusion pattern is blurred, which indicates the complete absence of crystallinity of Pt particles (Figure 1).

Apparently, a large number of surface defects and functional groups prevent the crystallization and aggregation of platinum nanoparticles during the formation of catalytically active planes at the stage of preparation of these catalysts. In order to prove that the investigated monometallic catalysts actually contain platinum, the samples were examined by the XPS method. The spectra of Pt 4f electrons manifest the main state of platinum (79% of the total content) in these catalysts to be characterized by a component with a binding energy of about 72 eV, which significantly exceeds the energy value of 71.1 eV, which is typical for bulk metallic platinum [31]. Such a difference in energies is usually associated with effects caused by small particle sizes, or with the transfer of electrons from platinum particles to a carbon carrier [32]. In the present work, the position of this component in the XP spectra did not change with changes in the size of platinum particles and the composition of catalysts (Figure 2, Appendix A). In this connection, we can assume that when platinum interacts with Sibunit, the formation of electron-deficient platinum Pt^δ+^ occurs, the particles of which form an incomplete polycrystalline structure.

On the other hand, for hydrogenation processes on Pt catalysts, it is known that the formation of Pt^δ+^ particles has a positive effect on the hydrogenation reaction, whereas, for the dehydrogenation reaction, the presence of uncharged platinum (Pt^0^) on the surface of the catalyst is more important [11]. At the same time, it follows from Table 1 and Table 3 that Pt/C_ox_ catalysts containing amorphous particles have high activity in the bicyclohexyl dehydrogenation reaction. So, the TOF of the catalyst 0.1Pt/C_ox_ approaches the specific activity of catalysts for the dehydrogenation of monocyclic hydrocarbons (cyclohexane, methylcyclohexane) with a much higher platinum content and significantly exceeds TOF in the case of bicyclic molecules—decalin (460 mmol(H_2_)/gPt × min). At the same time, the conversion of perhydro-*meta*-terphenyl on the 3Pt/C catalyst is higher than on Pt/Al_2_O_3_, even with the greater dispersion of Pt in the latter [33]. Taking into account the fact that electron-deficient platinum Pt^δ+^ is formed on aluminum oxide with a higher probability [11] than on carbon carriers, these facts seem to indicate a specific interaction of platinum and oxidized Sibunit. This may mean that the polycrystalline nature of platinum nanoparticles in a monometallic catalyst contributes to the emergence of many stable bonds with the functional groups of the carrier. This causes a high dispersion of the catalyst, and the stabilization of platinum particles leads to a decrease in their electron density and contributes to the high activity of single-component catalysts in the dehydrogenation reaction. The oxidation of Sibunit enhances these processes, which has a positive effect on the dehydrogenation of bicyclohexyl, compared with the untreated carrier.

From what has been said, it is quite obvious that in the case of Sibunit, an increase in the extent of functionalization of the surface of an inert carbon carrier and a decrease in diffusion restrictions play a more important role than a deterioration in its textural characteristics. At the same time, the results obtained indicate a limited possibility of oxidation, since the latter contributes to the degradation of the carrier. In this regard, in this work, to increase the catalytic activity of Pt nanoparticles, a method of small change in electron density was used by adding a second metal to platinum. The choice of modifying metals (Cr, Ni) is justified by the lower values of the electron work function for them in comparison with platinum (Table 4), due to which both metals in contact with Pt are presumably able to increase the electron density and catalytic activity of Pt nanoparticles. At the stage of preliminary studies, it was shown that the optimal concentrations are 3 wt.% for nickel and 1.5 wt.% for chromium [34,35,36]. The catalyst 0.1Pt/C_ox_ was used as the basic composition, whose TOF is significantly higher than that of other similar monometallic catalysts (Table 3).

Comparative analysis using XRD, SEM, and TEM data, including atomic resolution, electron diffraction, EDX mapping, XPS, in situ magnetometry, and TPR analysis, allowed us to establish a number of patterns indicating a change in the platinum activity in the bicyclohexyl dehydrogenation reaction in the presence of modifying metals. Comparison of platinum particle sizes in mono-, bi-, and trimetallic catalysts showed a correlation between d_Pt_ and bicyclohexyl conversion (Figure 3).

In contrast to the monometallic catalyst 0.1Pt/C_ox_, Pt particles in a bimetallic catalyst 0.1Pt/3Ni/C_ox_ have a completed crystal structure, and the characteristic size of metal particles increases to 10–20 nm [34]. The interaction of two metals with different redox properties leads to the oxidation of metallic nickel (NiO, 32%), compared with the monometallic catalyst 3Ni/C_ox_ (54%), as well as to a sharp decrease in the positive charge (δ+) on Pt nanoparticles and their transition mainly to the uncharged state Pt^0^ (91%) [36]. At the same time, the formation of solid solutions of Pt_x_Ni_1−x_ is observed on the surface of the catalyst [34], which creates geometric difficulties for the effective placement of active platinum atoms, compared with a Pt catalyst without nickel. The agglomeration of the metal and the change in the composition of the bimetallic nanoparticle is accompanied by a sharp drop in the conversion of bicyclohexyl in the dehydrogenation process.

In the CrPt system, as in the 0.1Pt/C catalyst, for example, a crystal lattice is also not observed on electron micrographs (Figure 4), but the Pt particles are large (d_Pt_ = 3–5 nm), which correlates with a decrease in the conversion of bicyclohexyl [34]. The latter can also be caused by the blocking of a part of the functional groups of oxidized Sibunit by chromium oxide when platinum is supported onto the oxidized surface of the substrate (1.5Cr/C_ox_) compared to the sample with metal supported on a carbon carrier. This reduces the degree of interaction of electron-deficient platinum particles Pt^δ+^ with the surface. Moreover, unlike the Pt-catalyst without chromium, the catalyst 0.1Pt/1.5Cr/C demonstrates an increase in the content of particles of electron-deficient platinum Pt^δ+^ (87%) [36], rather than its decrease, since the energy of the electron work function for chromium is lower than for nickel (Table 4). Apparently, in this case, platinum Pt^0^ reoxidation occurs when it interacts with acidic chromium oxide instead of metal. For the catalyst 0.1Pt/1.5Cr/C, as in the case of the corresponding platinum–nickel catalyst, this also leads to a decrease in the conversion of bicyclohexyl (X = 31%, S_(C12H10)_ = 67%, TOF = 115 mmol(H_2_)/g_Me_ * min) compared with a monometallic Pt/C_ox_ catalyst (*X* = 55%, *S*_(C12H10)_ = 68%, TOF = 1012 mmol(H_2_)/g_Me_ * min).

The comparison showed that a higher conversion of bicyclohexyl and selectivity for biphenyl was achieved on trimetallic PtCrNi systems than on a 0.1Pt/C_ox_ catalyst and regardless of the order of supporting the modifying metals [34]. At the same time, along with a decrease in the average size of Pt nanoparticles and an increase in the content of uncharged platinum (Pt^0^) by 1.5–2 times, a number of interesting facts were found indicating the existence of a mutual influence of metals on each other and their interaction with the carbon carrier. Thus, in the TPR curves of monometallic catalysts, deposition of Pt, Ni, or Cr on the carrier leads to a shift of the peak corresponding to the process of Sibunit hydrogenation towards lower temperatures (Appendix A), apparently due to the catalytic effect of the metal [36]. In the PtNi system, this peak practically disappears [35], which is apparently due to the blocking of the interaction of Pt and Ni particles with the carbon carrier in their simultaneous presence. This correlates with a sharp increase in the content of uncharged platinum in the XP spectra of the 0.1Pt/3Ni/C_ox_ catalyst. In the CrPt system, this peak is preserved on the TPR curves, as well as in the triple PtCrNi systems. Moreover, a decrease in the temperatures of all processes associated with the reduction in chromium and nickel is observed for the PtNiCr catalysts compared to the corresponding monometallic systems. Data from in situ magnetometry, XRD, and high-resolution TEM methods indicate the formation of chromium substitution solid solutions in nickel (Cr_x_Ni_1−x_) in the triple PtCrNi systems due to the strong interaction of Ni and Cr, which blocks the negative effect of nickel on platinum, compared with PtNi systems. Thus, the magnetometry method shows a decrease in the Curie temperature (T_C_) to 342–343 °C for three-component catalysts—0.1Pt/3Ni/1.5Cr/C_ox_ and 0.1Pt/(3Ni-1.5Cr)/C_ox_ and up to 323°C for the catalyst 0.1Pt/1.5Cr/3Ni/C_ox_, compared with (T_C_ =350 °C) for the monometallic nickel catalyst 3Ni/C_ox_ (Appendix A). A similar effect is known to occur on ferromagnetic metals when non-magnetic elements are introduced into their structure [34], in this case, chromium into nickel.

Diffractograms of a monometallic nickel catalyst 3Ni/C_ox_ (Figure 4) show the peak maxima at 2θ = 44–45° and 52° that corresponds to the crystallographic planes (111) and (200) of the FCC lattice of Ni [39] (2θ = 44.505° and 51.86°; PDF 4-850). Due to the low platinum content and the amorphous state of chromium, the diffractograms of the catalysts 0.1Pt/C_ox_ and 0.1Pt/1.5 Cr/C_ox_ were not considered, since the main catalytic effects are related to the position of nickel. The general identity of the lines with diffractograms of trimetallic PtNiCr catalysts indicates a very small amount of metallic chromium on the surface or its X-ray amorphous state. At the same time, there is a shift of the maxima of both Ni reflexes towards smaller angles, compared with the 3Ni/C_ox_ catalyst [34], which can also be explained by the formation of a solid substitution solution of chromium in nickel, since the simultaneous presence of metallic Ni and Cr in the system should widen the nickel reflex corresponding to the (111) plane. In particular, for the catalyst 0.1Pt/1.5Cr/3Ni/C_ox_ with the lowest T_C_, the largest increase in the lattice parameter is found (increasing from 0.3525 to 0.3541 nm). This is also evidenced by the data of high-resolution electron microscopy, in which the values of the inter-plane distances *d*_111_ and *d*_200_, calculated along the normal and confirmed by electron diffraction, also exceed those in the monometallic catalyst 3Ni/C_ox_.

It was shown in our previous studies that the solid solution formed on the surface of the studied catalysts contains regions with different chromium and nickel contents (Cr_x_Ni_1−x_). For example, for the investigated trimetallic catalysts, the chromium content varies from 1 to 10 at. % [34]. The calculation shows that for the formation of an average composition of Cr_0.022_Ni_0.978_ with the content of chromium 2.2 at. % in the catalyst 0.1Pt/1.5Cr/3Ni/C_ox_, about 50% of the total nickel content (3 wt.%) must be used. In particular, according to the stoichiometry of the catalyst 0.1%Pt-1.5%Cr-3%Ni/95.4%C_ox_, expressed, for example, in grams, about 0.033 g of chromium (1.5 × 2.2%) and 1.47 g of nickel are needed to achieve the composition of Cr_0.022_Ni_0.978_ (0.033 g × 97.8%). The latter is exactly half of the nickel in the catalyst. This allowed us to propose that, due to the surface segregation of Ni-Cr, the solid substitution solution forms local regions on the surface of triple PtCrNi systems, which play a key role in increasing the activity of trimetallic systems compared not only with bi-, but also with monometallic catalysts. In mono- and bimetallic catalysts 0.1Pt/C_ox_ and 0.1Pt/1.5Cr/C_ox_ deposited on oxidized Sibunit, high dispersion is achieved, but a high concentration of electron-deficient platinum Pt^δ+^ is formed, which is not always good for the dehydrogenation reaction [11]. When combined with nickel, a high concentration of non-oxidized platinum Pt^0^ is formed, which, in fact, is “poisoned” by nickel, including the effect of the formation of the Pt_x_Ni_1−x_ alloy. In the triple platinum–chromium–nickel system, the formation of highly stable Cr_x_Ni_1−x_ alloys inhibits the formation of Pt_x_Ni_1−x_ alloy, thereby significantly complicating the processes of deactivation of platinum particles by alloying with nickel and their agglomeration. A qualitative model of the effect of the interaction of metals with different redox properties on the process of dehydrogenation of bicyclohexyl based on the catalytic results and data of physico-chemical analysis of mono-, bi-, and trimetallic catalysts is shown in Figure 5.

## 3. Materials and Methods

### 3.1. Preparation of Catalysts

Mono-, bi-, and trimetallic catalysts deposited on the carbon carrier Sibunit^®^ (C, Omsk, Russia, bulk density 0.62 g/cm^3^, average granule diameter of 1.5–1.8 mm, a specific surface area of 243 m^2^/g, an average pore size of 7.3 nm and the pore volume of 0.45 cm^3^/g) were used in this work. For comparison, similar catalysts prepared using oxidized Sibunit (C_ox_) were used [16]. The composition and conditions for the preparation of catalysts are given in Table 5.

### 3.2. Characterization of Catalysts

The particle size (R) and dispersion (D) of platinum were determined using the ASAP 2020 microanalyzer (Norcross, GA, USA) by irreversible CO chemisorption at 35 °C. The surface morphology of the catalysts was studied by transmission electron microscopy (TEM) with a JEOL-2100F (Tokyo, Japan) electron microscope in light and dark field modes at an accelerating voltage of 200 kV.

The charge state of metals and the composition of the catalyst surface were determined by X-ray photoelectron spectroscopy (XPS) using a Kratos Axis Ultra DLD device (Great Britain) with monochromatic radiation Al Ka (hυ = 1486.6 eV, 150 W). The standard energy of the analyzer was 160 eV and 40 eV for high-resolution spectra.

The phase composition of the samples was studied by X-ray phase analysis (XRD) using an automatic diffractometer DRON–3 (Russia) with CuKa radiation (λ = 1.5405 A) with a graphite monochromator in step-by-step scanning mode (step 0.1°, exposure per point—5 s). The main scanning interval is 20–90°.

The dependence of the rate of hydrogen absorption in the mode of temperature-programmed reduction (TPR) of catalysts was studied using a laboratory plug-flow installation KL-1 (Moscow, Russia). The reduction was carried out with a mixture of gases of 5% H_2_/Ar at a flow rate of 23 mL/min to a temperature of 700 °C. The rate of linear heating of the detector was 10 °C/min.

Chemical transformations in catalysts during the reduction process were investigated by the magnetometric method under in situ conditions [34].

### 3.3. Dehydrogenation of Bicyclohexyl

The dehydrogenation reaction was carried out in a flow reactor. Samples of all the studied catalysts with a weight of 1.85 g were placed in a steel reactor with a diameter of 10 mm and activated in accordance with the above procedures. After that, the temperature was brought to the reaction temperature and the substrate was fed with a high-pressure pump HPR 5001 with a linear feed rate of 6 mL/h (density 0.864 g/cm^3^). Commercial bicyclohexyl (99%, Acros Organics, Geel, Belgium, C_12_H_22_) was used as a substrate. Dehydrogenation was carried out at a temperature of 320 °C and atmospheric pressure for 4 h. Hydrogen and reaction products were separated.

The reaction products were analyzed using a Crystallux-4000M chromatograph (Kazan, Russia) with a ZB-5 capillary column (Zebron Phenomenex, Torrance, CA, USA) and a flame ionization detector of the FOCUS DSQ II chromatography–mass spectrometer (Thermo Fisher Scientific, Waltham, MA, USA) with a TR-5ms capillary column. The analysis was performed in a programmable temperature mode of 70–220 °C at a heating rate of 6 °C/min.

The conversion (X) of bicyclohexyl was calculated as the ratio of the change in the amount of bicyclohexyl before and after the reaction to the initial amount of bicyclohexyl. The selectivity (S) to the reaction products was determined as the ratio of the formed amount of one of the reaction products to the total amount of products. The experimental error in determining the conversion and selectivity values was not larger than 3 rel. %.

## 4. Conclusions

Thus, the increase in the conversion of bicyclohexyl and selectivity for biphenyl in the dehydrogenation reaction on trimetallic catalysts is due to an increase in the activity of platinum in the vicinity of local accumulations of Cr-Ni solid solutions, in which the intensity of electron transfer from platinum to nickel is lower than in catalysts without chromium. The high activity of catalysts containing no Cr-Ni solid solution depends on the morphology and dispersion (particle size) of metals, as well as on their electronic interaction between themselves and with the carrier.

## Figures and Tables

**Figure 1 molecules-27-08416-f001:**
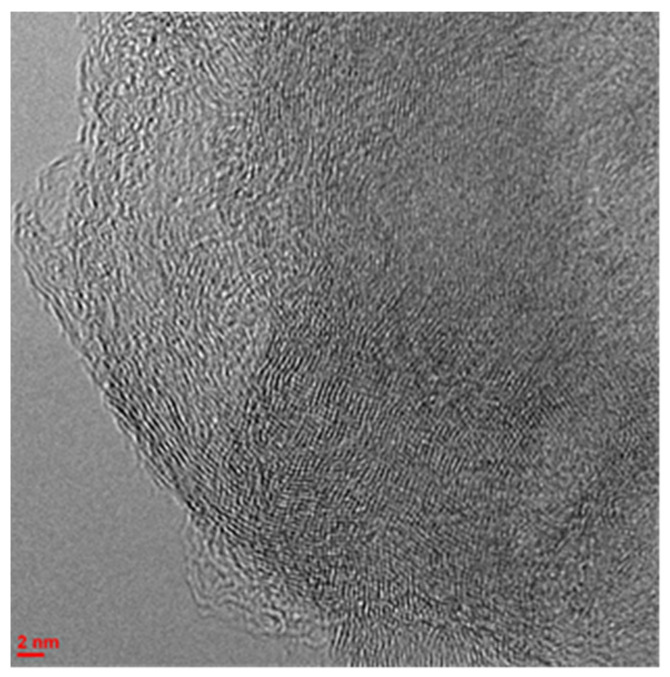
Micrograph of the surface of the catalyst 0.1Pt/C_ox_.

**Figure 2 molecules-27-08416-f002:**
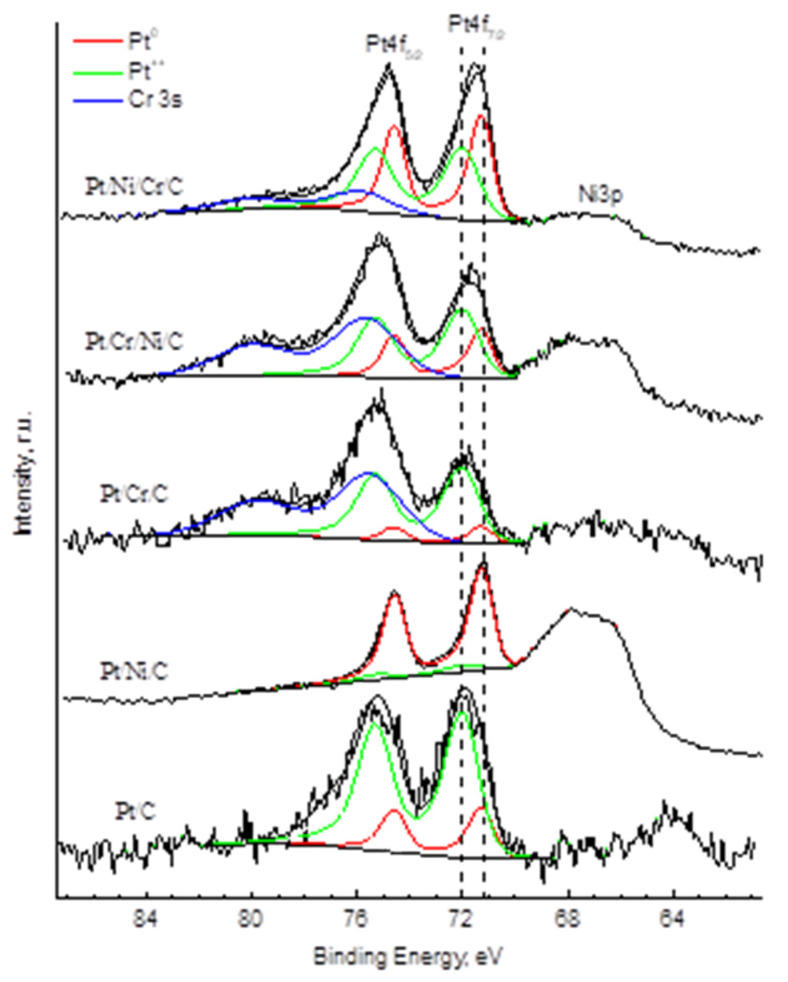
XP spectra of Pt 4f and Cr 3s electrons of Pt/C catalysts (C=C_ox_; Pt—0.1 wt.%) after being reduced in H_2_.

**Figure 3 molecules-27-08416-f003:**
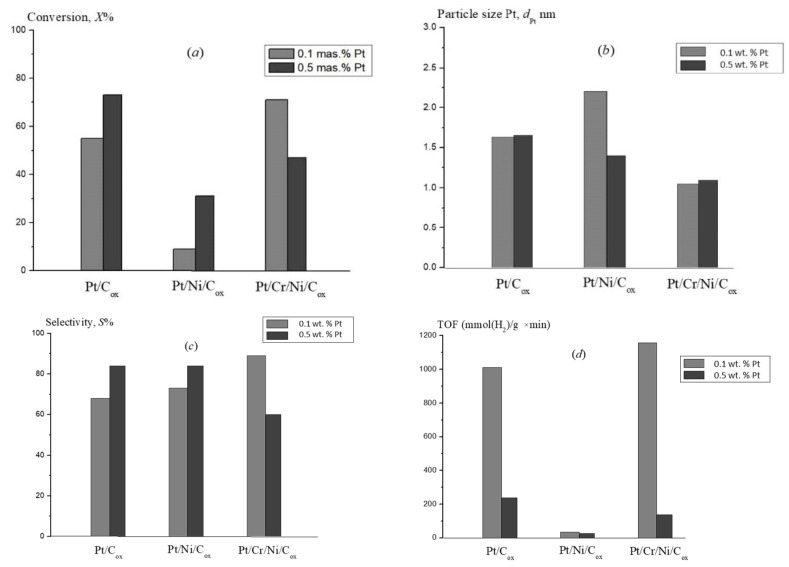
Bicyclohexyl (**a**) dehydrogenation conversions, (**b**) average Pt particle sizes, (**c**) selectivity to biphenyl, and (**d**) TOF values for mono-, bi-, and trimetallic catalysts with a Pt content of 0.1 and 0.5 wt.%.

**Figure 4 molecules-27-08416-f004:**
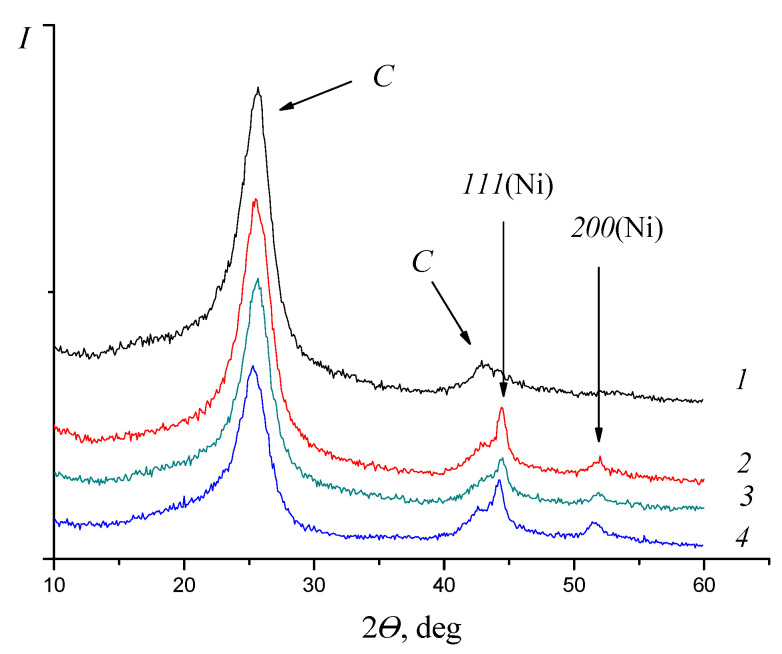
Diffraction patterns of C_ox_ and Ni-catalysts: 1—C; 2—Ni/C; 3—Pt/Ni/Cr/C; 4—Pt/Cr/Ni/C.

**Figure 5 molecules-27-08416-f005:**
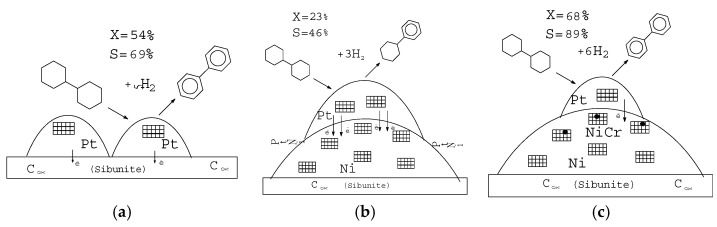
Scheme of bicyclohexyl dehydrogenation on catalysts 0.1Pt/C_ox_ (**a**), 0.1Pt/3Ni/C_ox_ (**b**) and 0.1Pt/1.5Cr/3Ni/C_ox_ (**c**).

**Table 1 molecules-27-08416-t001:** Hydrogen release rates during dehydrogenation of some cycloalkanes.

Substrate	Catalyst	TOF, (mmol(H_2_)/g_Pt_ × min)	Ref.
Cyclohexane	3.82 wt% Pt/AC (activated carbon)	1800	[2]
Methylcyclohexane	3.82 wt% Pt/AC	1700	[2]
Decalin	5 wt% Pt/C	444.4	[2]
Decalin	(Pt-Ir)/C (5 wt% metal)	25.2	[2]
Perhydro-*meta*-terphenyl	5Pt/Al_2_O_3_	7.1	[17]
Perhydro-*meta*-terphenyl	4.8Pd/C	15.1	[17]
Perhydro-*meta*-terphenyl	3Pt/C(Sibunit)	56.2	[17]

**Table 2 molecules-27-08416-t002:** Textural characteristics of the carrier and the proportion of carbon atoms in various chemical states (%) according to the XPS data (C—Sibunit; C_ox_—oxidized Sibunit).

	S_BET_, m^2^/g	V_pore_, cm^3^/g	r_pore_, nm	C-C (sp^2^)	C-C (sp^3^)	C–O	C=O	O=C-O
C	360	0.55	3.8	~98%	-	-	-	1–2%
C_ox_	240	0.45	4.2	~88%	5–7%	3–4%	0–1%	3–4%

**Table 3 molecules-27-08416-t003:** Activity of Pt/C catalysts in the bicyclohexyl dehydrogenation reaction (T = 320 °C, *p* = 1 atm, VHSV = 2.8 h^−1^).

Parameter	Catalyst					
	0.5Pt/C	0.5Pt/C_ox_	0.3Pt/C	0.3Pt/C_ox_	0.1Pt/C	0.1Pt/C_ox_
*X*_C12H22_, % (±5%)	63	73	57	64	26	55
*S*_C12H10_, %	61	84	75	79	58	68
TOF, (mmol(H_2_)/gPt × min)	184	238	287	359	855	1012

**Table 4 molecules-27-08416-t004:** Work function and surface energy of metallic Pt, Ni, and Cr.

Metal	Work Function, eV [37]	Surface Energy, J/m^2^ [38]
Pt	5.93	2.067
Ni	5.04	2.450
Cr	4.50	2.300

**Table 5 molecules-27-08416-t005:** Conditions for the preparation of catalysts.

Catalyst	The Sequence of Preparation Stages
3Ni/C; 3Ni/1.5Cr/C	Impregnation (C; 1.5Cr/C) with an aqueous solution of Ni(NO_3_)_2_ 6H_2_O (Sigma Aldrich), drying (120 °C), calcination in N_2_ (500 °C), reduction of Ni in H_2_ (500 °C)
1.5Cr/C 1.5Cr/3Ni/C	Impregnation (C; 3Ni/C) with an aqueous solution of Cr(NO_3_)_3_ 9H_2_O (Sigma Aldrich, Saint-Louis, MO, USA), drying (120 °C), calcination in N_2_ (500 °C), reduction of Cr in H_2_ (500 °C)
(1.5Cr-3Ni)/C	Co-impregnation (C) with aqueous solutions of Ni(NO_3_)_2_ 6H_2_O and Cr(NO_3_)_3_ 9H_2_O, drying (120 °C), calcination in N_2_ (500 °C), reduction of Ni and Cr in H_2_ (500 °C)
0.1Pt/C; 0.1Pt/3Ni/C; 0.1Pt/1.5Cr/C; 0.1Pt//3Ni/1.5Cr/C; 0.1Pt/1.5Cr/3Ni/C; 0.1Pt/(1.5Cr-3Ni)/C	Impregnation (C; 3Ni/C; 1.5Cr/C; 3Ni/1.5Cr/C; 1.5Cr/3Ni/C; (1.5Cr-3Ni)/C) with an aqueous solution of H_2_PtCl_6_ 6H_2_O (Alfa-Aesar, Stoughton, MA, USA), drying (120 °C), calcination in N_2_ (350 °C), reduction of Pt in H_2_ (320 °C)

## Data Availability

Not applicable.

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
