# Peer review of "Activity of Mono-, Bi-, and Trimetallic Catalysts Pt-Ni-Cr/C in the Bicyclohexyl Dehydrogenation Reaction"

_molecules, 2022, doi:10.3390/molecules27238416_

Round 1
Reviewer 1 Report
The work by Kalenchuk et al. details a study on evaluating bicyclohexyl dehydrogenation using mono, bi-, and trimetallic catalysts Pt-Ni-Cr/C. The catalysts are characterized by specific surface area (BET method), crystallinity (XRD), morphology (HR-TEM), and chemical composition (XPS). However, it is not clear the relationship between the catalyst’s properties and the resulting catalytic performance. Therefore, I recommend publication of the present manuscript in the Molecules (MDPI) Journal after minor revision. Additional comments are provided attached.

Author Response
We would like to thank the reviewer for the critical comments and useful advices and questions raised. We tried to do our best to improve our manuscript by taking into account all the comments. Below we give our responses to the comments. The changes in the text are highlighted in yellow.
Comments and Suggestions for Authors
The work by Kalenchuk et al. details a study on evaluating bicyclohexyl dehydrogenation using mono, bi-, and trimetallic catalysts Pt-Ni-Cr/C. The catalysts are characterized by specific surface area (BET method), crystallinity (XRD), morphology (HR-TEM), and chemical composition (XPS). However, it is not clear the relationship between the catalyst’s properties and the resulting catalytic performance. Therefore, I recommend publication of the present manuscript in the Molecules (MDPI) Journal after minor revision. Additional comments are provided attached.
Response: In the paper, much attention is paid to the formation of electron-deficient platinum (Ptδ+), the amount of which, compared with metallic platinum (Pt0), affects the overall activity of the catalyst. We have shown that the contact of functional groups of oxidized sibunit as electron acceptor centers with nanoparticles of electron-deficient platinum has a stabilizing effect on the latter, which leads to an increase in dispersion and suppression of agglomeration and, accordingly, to an increase in the activity of platinum catalysts.
Abstract & Introduction
- See abstract. It might be extended. Moreover, it is essential to include the most relevant experimental data.
Response: The abstract text is expanded by comparing the most important experimental data:
“An increase in the conversion (X>62%) of bicyclohexyl and selectivity for biphenyl (S > 84%) was observed on trimetallic catalysts Pt-Ni-Cr/C compared with the monometallic catalyst Rt/C (X> 55%; S >68%).”
Materials and Methods
- All of the materials should include brand name (i.e. Sigma Aldrich, Merck, etc.) and purity (in percentage).
Response: The necessary changes have been made in the text.
Results and Discussion
- It is necessary to see the Cr2p core level from XPS measurement to observe the ratio of the different oxidation states of Cr. In addition, from Ni3p or Ni2p core levels XPS spectra, it should distinguish between different oxidation states of Ni.
Response: Since platinum is the main active metal in the studied mono-, bi- and trimetallic systems, the data presented in the article are primarily aimed at showing an important role in increasing its activity in trimetallic systems under the influence of modifying components, such as Ni and Cr. At the same time, we did not consider the influence of nickel and platinum on chromium in this work. Nevertheless, Supplementary Information (Table S3) shows the characteristics of the XPS spectra of Ni-catalysts after reduction in H2. At the same time, the Cr2p XPS spectra of samples of all the studied catalysts after reduction in hydrogen are observed as a doublet of wide lines with a poorly resolved structure and with a binding energy of the Cr2p3/2 component equal to 577.0 eV, which is typical for trivalent chromium compounds. Traces of metallic chromium in these samples are not detected even taking into account the reduction treatment, apparently due to its low content in the catalysts. For some samples, we observed a small difference in the binding energy (576.8 eV). This may be due to the partial reduction of chromium oxide occurring at the boundary of the oxide and carbon phases of sibunit during calcination of catalyst samples at 500oC even in the absence of hydrogen (for instance, via the reaction 2CrO +C = 2Cr + CO2).
- Is there any correlation between the oxidation state of metals, such as Ni, Pt, and Cr (based on the XPS, Figure 2) with the catalytic performance (Figure 3)?
Response: In the mono-, bi- and trimetallic catalysts studied in this work, Ni and Cr exhibit weak activity in the bicyclohexyl dehydrogenation reaction. The main active metal is platinum, whose interaction with sibunit leads to the formation of a large number of electron-deficient Pt(δ+) particles. We found that the catalytic characteristics of this reaction depend on the ratio of the content of metallic Pt(0) and electron-deficient platinum particles Pt(δ+) in the catalyst.
- HR-TEM of bimetallic and trimetallic catalysts should be added in the revised manuscript.
Response: The most typical examples of HR-TEM for bimetallic and trimetallic catalysts are presented in the Supplementary Information.
- The diffractogram of the XRD pattern should be added in the revised manuscript.
Response: Diffractograms of the studied catalysts are presented in the Supplementary Information.
- The catalytic performance of PtCr/Cox should be added in the revised manuscript.
Response: The catalytic characteristics obtained on PtCr/Cox have been added to the text:
“As in the case of nickel, this also leads to a decrease in the conversion of bicyclohexyl (X = 31%), compared with a monometallic Pt/Cox catalyst.”
- Why PtNi/Cox shows significantly lower activity compared to monometallic Pt/Cox?
Response: In a single-component platinum catalyst deposited on sibunit, high dispersion is achieved, and the resulting particles of electron-deficient platinum are stabilized on the surface due to specific interaction with the carbon carrier sibunit, which positively affects the dehydrogenation reaction of bicyclohexyl. When combined with nickel, a large portion of non-oxidized platinum is formed, but which, in fact, is "poisoned" by nickel due to the binding of Pt in the form of solid solutions of PtxNi1-x, which create geometric difficulties for the effective placement of active platinum atoms, compared with a Pt catalyst without nickel. The agglomeration of the metal and the change in the composition of the bimetallic nanoparticle is accompanied by a sharp drop in the conversion in bicyclohexyl dehydrogenation.
- Based on Figure 3b, how many Pt particles are counted for each catalyst? At least 50-100 particles to make sure that the average size is appropriate.
Response: To calculate the average size, at least 100 platinum particles were taken using several micrographs of TEM, including HR TEM.
- To propose the brief mechanism should be based on the physiochemical characteristics of catalysts. What is the basis of the construction geometry of metals illustrated in Figure 3b? Was it based on the HR-TEM/other characterizations or previous literatures?
Response: An important point of this work is that the necessary effect occurs on a catalyst with a low platinum content, which is achieved through promotion. In a single-component platinum catalyst deposited on sibunit, high dispersion is achieved, but a large portion of electron-deficient platinum Pt(δ+) species is formed, which is not always good for the dehydrogenation reaction. However, the functional groups of oxidized sibunit act as electron acceptor centers, interaction with them leads to a decrease in the electron density on the Pt nanoparticle and to stabilization of Pt(δ+) particles on the surface of single-component Pt/C systems, which has a positive effect on its activity in the bicyclohexyl dehydrogenation reaction.
When two metals with different redox properties interact, in the case of nickel, metal agglomeration and a change in the composition of the bimetallic nanoparticles occur, which negatively affect the catalytic properties of PtNi/C systems and is accompanied by a sharp drop in the conversion in bicyclohexyl dehydrogenation, compared with Pt/C, despite the high content of non-oxidized platinum Pt(0) species. In the case of chromium, in PtCr/C, an increase in the amount of electron-deficient platinum Pt(δ+) is observed, the stabilization of particles of which is reduced when platinum is supported onto the oxidized surface of the substrate Cr/C, compared with the carbon carrier in the Pt/C system.
In the triple platinum-chromium-nickel system, the interaction of metals leads to an increase in the content of non-oxidized platinum, which contributes to an increase in the conversion of bicyclohexyl. To establish a more precise mechanism, a much more thorough studies of metal-metal contacts is required, which is the subject of further research by the authors.
- Grammar and, in particular, sentence structure in several cases need to be improved within the manuscript so that the authors' point is clear. Please kindly check all grammatical/typo errors in the manuscript.
Response: The necessary changes have been made in the text.
- The Graphical abstract and proposed mechanism must effectively capture the essence of the study.
Response: The graphic abstract is presented more accurately.
Reviewer 2 Report
The manuscript entitled “Activity of mono-, bi- and trimetallic catalysts Pt-Ni-Cr/C in the bicyclohexyl dehydrogenation reaction” reports research results regarding the synthesis, characterization of catalyst for bicyclohexyl dehydrogenation reaction as the release stage of hydrogen in hydrogen storage. The research results are essential contributions to the existing scientific knowledge in the area of biofuel production using heterogeneous catalysts. Although the manuscript presents a series of specific analyses, it still needs modification before its acceptance in the Journal Molecules. The revised version of the manuscript may be submitted taking into account the following comments before final acceptance:
In lines 70-71, the article describes that during oxidation, the textural deterioration of Sibunite occurred, leading to a reduction in the specific surface area (SBET) and in the total volume of pores. As the article also applies BET to confirm this conclusion and measure the surface and pore size of these catalysts. The addition of nitrogen physisorption isotherms would improve the reading of these data by readers. That said, what could have caused the textural properties to change? Justify in manuscript.
Check if it is correctly referring to (Table 3) or should be (Table 4) Line 175. The manuscript needs a thorough review mainly in the acronyms
The Pt/Cr/C catalyst results should be added to Figures (3a and 3b) for readers to clearly visualize the information. Furthermore, in Figure 3 there should be results such as Selectivity and TOF obtained by the catalysts. Although the catalysts studied in the dehydrogenation of bicyclohexyl are shown in Figure 3. Catalyst characterization results are used to discuss the conversion of catalytic results. However, there is a lack of discussion about the role of catalytic active sites, such as functions of metallic particles on selectivity to desired products.
The addition of TPR curves, XPS spectra, XRD and TEM catalysts would improve the manuscript so that readers can easily follow the text and data to be analyzed. It is easy to read that the manuscript can add these data in a table and compare the parameters analyzed as well reported in other works.
The manuscript describes an investigation of chemical transformations in catalysts during the reduction process by the magnetometric method would improve if the magnetization temperature dependence curves and the magnetization curves for catalysts for the synthesized catalysts were included in the manuscript.
The manuscript describes that the triple platinum-chromium-nickel system leads to the formation of highly stable CrxNi1-x alloys by inhibiting the formation of the PtxNi1-x alloy, significantly complicating the processes of deactivation of platinum particles by alloying with nickel and their agglomeration. If catalyst deactivation occurs during the reaction, it is important that these catalysts are regenerated and reused. Recycling tests should show that leaching is not contributing to final decommissioning, with an emphasis on material composition and possible stabilities. Furthermore, the characterization of reused catalysts, in most cases, should help to explain such results. Currently, this is not the case, as it is difficult to follow the stability related to the composition of the reported materials . Therefore, taking into account that the study has only one reaction test result for each material (Figure 3), which is not sufficient to determine the catalytic effect, or to establish any correlation with the composition of the catalyst. Thus, a study of recycling and recharacterization of the catalysts should be carried out to improve the manuscript.

Author Response
We would like to thank the reviewer for the critical comments and useful advices and questions raised. We tried to do our best to improve our manuscript by taking into account all the comments. Below we give our responses to the comments. The changes in the text are highlighted in yellow.
Comments and Suggestions for Authors
The manuscript entitled “Activity of mono-, bi- and trimetallic catalysts Pt-Ni-Cr/C in the bicyclohexyl dehydrogenation reaction” reports research results regarding the synthesis, characterization of catalyst for bicyclohexyl dehydrogenation reaction as the release stage of hydrogen in hydrogen storage. The research results are essential contributions to the existing scientific knowledge in the area of biofuel production using heterogeneous catalysts. Although the manuscript presents a series of specific analyses, it still needs modification before its acceptance in the Journal Molecules. The revised version of the manuscript may be submitted taking into account the following comments before final acceptance.
- In lines 70-71, the article describes that during oxidation, the textural deterioration of Sibunit occurred, leading to a reduction in the specific surface area (SBET) and in the total volume of pores. As the article also applies BET to confirm this conclusion and measure the surface and pore size of these catalysts. The addition of nitrogen physisorption isotherms would improve the reading of these data by readers. That said, what could have caused the textural properties to change? Justify in manuscript.
Response: In this study, the change in the surface area did not have a significant effect on the activity of three-component catalysts. The increase in activity occurred with a decrease in the surface area, although it would seem that it should have been the other way around. The decrease in the surface area during the oxidation of the carbon carrier occurred due to the partial destruction of the inner walls of the pores, but this also led to an increase in the pore size, which contributed to a decrease in diffusion restrictions during the dehydrogenation of such a large molecule as bicyclohexyl and, as a consequence, to an increase in the catalyst accessibility on the oxidized carrier. The spectra of the distribution of pore sizes in the carrier before and after oxidation according to CO chemisorption data are given in Supplementary Information (Fig. S1).
- Check if it is correctly referring to (Table 3) or should be (Table 4) Line 175. The manuscript needs a thorough review mainly in the acronyms
Response: Some typos in Table 2 (Vpore and rpore ) were fixed. In the text (Line 175), the number of "Table 3" has been corrected to "Table 4".
- The Pt/Cr/C catalyst results should be added to Figures (3a and 3b) for readers to clearly visualize the information. Furthermore, in Figure 3 there should be results such as Selectivity and TOF obtained by the catalysts. Although the catalysts studied in the dehydrogenation of bicyclohexyl are shown in Figure 3. Catalyst characterization results are used to discuss the conversion of catalytic results. However, there is a lack of discussion about the role of catalytic active sites, such as functions of metallic particles on selectivity to desired products.
Response: а) Diagrams with values of selectivity for biphenyl (c) and TOF (d) for mono-, bi- and trimetallic catalysts have been added to the diagrams (Fig. 3) with values of conversion (a) and particle sizes (b).
- b) The work pays great attention to the formation of electron-deficient platinum, the ratio of the content of which to metallic platinum affects the overall activity of the catalyst. Wers have shown that the contact of functional groups of oxidized sibunit as electron acceptor centers with nanoparticles of electron-deficient platinum has a stabilizing effect on the latter, which leads to an increase in dispersion and suppression of agglomeration and, accordingly, to an increase in the activity of platinum catalysts.
- The addition of TPR curves, XPS spectra, XRD and TEM catalysts would improve the manuscript so that readers can easily follow the text and data to be analyzed. It is easy to read that the manuscript can add these data in a table and compare the parameters analyzed as well reported in other works.
Response: a) XPS spectra of the compared catalysts are given in the text of the article (Fig. 2).
- b) At a low Pt content in the catalysts, platinum peaks do not appear on the corresponding TPR curves. The same applies to chromium, whose reduction rate in hydrogen is extremely low. Supplementary Information (Table S1) shows peak temperatures for comparison catalysts that do not relate to this article, but on the basis of which conclusions were drawn for the catalysts under study. There is a link in the text [35], where some data from this table are given.
- c) Due to the low platinum content and the amorphous state of chromium, adapted diffractograms of Ni-catalysts were given in Supplementary Information (Fig. S2), and the studied catalysts were compared. There is a link in the text [34] where this figure is used. The diffractogram of the 0.1Pt/C catalyst was not taken into account, since the main catalytic effects are related to the position of nickel.
- d) Supplementary Information shows the TEM of the catalyst 0.1Pt/C (a), which shows the uniform distribution of the metal, as well as the high-resolution TEM of the catalysts compared in the text 0.1Pt/3Ni/S (b) 0.1Pt/1.5Cr/3Ni/S (c).
5.The manuscript describes an investigation of chemical transformations in catalysts during the reduction process by the magnetometric method would improve if the magnetization temperature dependence curves and the magnetization curves for catalysts for the synthesized catalysts were included in the manuscript.
Response: The magnetization data are given in Supplementary Information (Table S2). There is a link in the text [34], where some data from this table are presented.
- The manuscript describes that the triple platinum-chromium-nickel system leads to the formation of highly stable CrxNi1-x alloys by inhibiting the formation of the PtxNi1-x alloy, significantly complicating the processes of deactivation of platinum particles by alloying with nickel and their agglomeration. If catalyst deactivation occurs during the reaction, it is important that these catalysts are regenerated and reused. Recycling tests should show that leaching is not contributing to final decommissioning, with an emphasis on material composition and possible stabilities. Furthermore, the characterization of reused catalysts, in most cases, should help to explain such results. Currently, this is not the case, as it is difficult to follow the stability related to the composition of the reported materials. Therefore, taking into account that the study has only one reaction test result for each material (Figure 3), which is not sufficient to determine the catalytic effect, or to establish any correlation with the composition of the catalyst. Thus, a study of recycling and recharacterization of the catalysts should be carried out to improve the manuscript.
Response: This paper does not consider deactivation of the catalyst as such, but a decrease in the activity of platinum when interacting with nickel. Since the optimal composition of the catalyst is in the stage of continuous optimization, no special studies have been conducted on the recycling of the catalyst and the analysis of catalysts after several cycles of use. Instead, the very good stability of the tested catalysts was observed for at least 8 hours, and for some samples - for at least 20 hours of continuous operation in the bicyclohexyl dehydrogenation reaction under optimal conditions was carried out at least 3 times on each of the studied catalysts. The text provides the average results. So no regeneration is required and no recyclization or leaching can be discussed for such stable catalysts.
Reviewer 3 Report
In this work, surface effects during the formation of mono-, bi- and trimetallic Pt—Ni—Cr catalysts were studied by CO chemical absorption in the reaction of bicyclohexyl dehydrogenation. The activity of Pt nanoparticles was regulated to an optimal level by Ni-Cr clusters, and the binding energy of Pt was clearly distinguished by XPS spectroscopy. However, the choice of bicyclohexyl as the hydrogen carrier does not seem to be very effective. Overall, this is a well-founded conclusion and oriented towards practical applications. I would like to recommend its acceptance for publication after the following minor revisions:
1. A TEM image containing nanoparticles should be given, and the size range of the particles included.
2. Please provide mapping images of different elements in the nanoparticles to check if the elements are evenly distributed.
3. Please provide the XRD patterns of the nanoparticles.
4. In Figure 2, please split the XPS spectra of Ni 3p.
5. The peak of Cr 3s is very different in Pt/Ni/Cr/C and Pt/Cr/Ni/C, please recheck the peak splitting parameters.
Author Response
We would like to thank the reviewer for the critical comments and useful advices and questions raised. We tried to do our best to improve our manuscript by taking into account all the comments. Below we give our responses to the comments. The changes in the text are highlighted in yellow.
Comments and Suggestions for Authors
In this work, surface effects during the formation of mono-, bi- and trimetallic Pt—Ni—Cr catalysts were studied by CO chemical absorption in the reaction of bicyclohexyl dehydrogenation. The activity of Pt nanoparticles was regulated to an optimal level by Ni-Cr clusters, and the binding energy of Pt was clearly distinguished by XPS spectroscopy. However, the choice of bicyclohexyl as the hydrogen carrier does not seem to be very effective. Overall, this is a well-founded conclusion and oriented towards practical applications. I would like to recommend its acceptance for publication after the following minor revisions.
Response: In a previous article in Molecules, the authors showed that bicyclohexyl is one of the polycyclic substrates, which is suitable for hydrogen storage in terms of reaction kinetics. In this article, we investigate the reason for the high efficiency of the bicyclohexyl dehydrogenation catalyst.
- A TEM image containing nanoparticles should be given, and the size range of the particles included.
Response: Examples of HR TEM showing nanoparticles with an indication of the particle size range are given in Supplementary Information (Fig. S3 b,c).
- Please, provide mapping images of different elements in the nanoparticles to check if the elements are evenly distributed.
Response: An example of a TEM image for the 0.1Pt/C catalyst showing a uniform distribution of platinum over the surface is given in Supplementary Information (Fig. S3a).
- Please, provide the XRD patterns of the nanoparticles.
Response: Examples of diffractograms of the studied catalysts are given in Supplementary Information (Fig. S2).
- In Figure 2, please split the XPS spectra of Ni 3p.
Response: Supplementary Information (Table S3) shows the characteristics of the XPS spectra of Ni-catalysts after reduction in H2. There is a link in the text [35], where some data from this table are given.
- The peak of Cr 3s is very different in Pt/Ni/Cr/C and Pt/Cr/Ni/C, please, recheck the peak splitting parameters.
Response: Cr2p XPS spectra of samples of all studied catalysts after reduction in hydrogen are observed as a doublet of wide lines with poorly defined structure and with a binding energy of the Cr2p3/2 component equal to 577.0 eV, which is typical for trivalent chromium compounds. Traces of metallic chromium in these samples are not detected even taking into account the reduction treatment, apparently due to its low content in catalysts. In addition, the XPS method provides information on the thickness of the layer up to 30 Å and metallic chromium cannot be detected due to the fact that due to migration it may be in the bulk of the sample, and not on the surface, for example, in the pores of the carrier. At the same time, for some samples, we observed a small difference in energy (576.8 eV). This may be due to the partial reduction of chromium oxide occurring at the boundary of the oxide and carbon phases of sibunit during calcination of catalyst samples at 500oC even in the absence of hydrogen (for instance, via the reaction 2Cr +C = 2Cr + CO2). Since the main attention is paid to the activity of platinum, we did not consider the influence of nickel and platinum on chromium in this work.
Round 2
Reviewer 2 Report
I recommend publishing this manuscript in the Molecules (MDPI) Journal after minor review. Additional comments are provided in the annex.

Author Response
We would like to thank the reviewer for the critical comments and useful advices and questions raised. We tried to do our best to further improve our manuscript by taking into account all the comments. Below we give our responses to the comments. The changes in the text are highlighted in yellow.
Review report 2
I recommend publishing this manuscript in the Molecules (MDPI) Journal after minor review. Additional comments are provided in the annex. The language of the manuscript has significantly improved over the previous version, and the article now has a good read. Showing that there was a revision in the previous version of the manuscript and it was significantly improved. I see my previous comments addressed and may suggest publishing the manuscript taking into account the following comments.
Please add to Figures (3a, 3b, 3c and 3d) the catalytic performance results of the PtCr/Cox catalyst (dehydrogenation conversions, average Pt particle sizes, selectivity to biphenyl and TOF) so that readers can clearly see this information. And mainly to reinforce the discussion: lines 182-184 “In the CrPt system, as in the 0.1Pt/C catalyst, for example, a crystal lattice is also not observed on electron micrographs, but the Pt particles themselves are large in size, which correlates with a decrease in the conversion of bicyclohexyl”.
Response: The effects found by the authors of this study were more pronounced on catalysts with a platinum content of 0.1 wt.%. Figures (3a, 3b, 3c and 3d) present data for catalysts with a platinum content of 0.1 wt.% in comparison with catalysts with a platinum content of 0.5 wt.%. For PtCr/Cox catalysts, the corresponding parameters are obtained only for the Pt content of 0.5 wt. %. We did not obtain the catalytic data for all systems, therefore, the graphical representation of incomplete data in one figure is incorrect, since it can cause no less reasonable questions and comments from both reviewers and editors, as well as readers. In this regard, the necessary information for the 0.1Pt1.5Cr/Cox catalyst under study is given in the text of the article. The following sentence has been added: “For the catalyst 0.1Pt/1.5Cr/C, as in the case of the corresponding platinum-nickel catalyst, this also leads to a decrease in the conversion of bicyclohexyl (X=31%, S(C12H10)=67%, TOF= 115 mmol(H2)/gМе * min) compared with a monometallic Pt/Cox catalyst (X=55%, S(C12H10)=68%, TOF= 1012 mmol(H2)/gМе * min).”
Also, the following sentences has been revised as follows:
“In the CrPt system, as in the 0.1Pt/C catalyst, for example, a crystal lattice is also not observed on electron micrographs (Fig. 4), but the Pt particles are large (dPt=3-5 nm), which correlates with a decrease in the conversion of bicyclohexyl [34].”
The TEM image of the 0.1Pt/1.5Cr/Cox catalyst was also added to Fig. S2.
Still in Figure 3, the manuscript investigates dehydrogenation conversions, average Pt particle sizes, selectivity to biphenyl and TOF. If the text gives average results, I suggest adding an error bar to the bar graphs.
Response: The text has been revised and the experimental error (3 rel. %) was indicated in the text in order to avoid complication of the figure.
Please, I suggest that the diffractograms presented in Complementary Information (Fig. S2) be added to the manuscript. And that the discussions placed in the answer are better elaborated and added in the manuscript “Due to the low content of platinum and the amorphous state of chromium, adapted diffractograms of Ni catalysts were presented in Complementary Information (Fig. S2), and the catalysts studied were compared. The 0.1Pt/C catalyst diffractogram was not taken into account, as the main catalytic effects are related to the nickel position”.
Response: The diffraction patterns were transferred from Supplementary Information to the main document (Fig. 4). The reference to the XRD pattern was added (Fig. 4). Also, the following phrase was added on p. 7: “Due to the low platinum content and the amorphous state of chromium, the diffracto-grams of the catalysts 0.1Pt/Cox and 0.1Pt/1.5 Cr/Cox were not considered, since the main catalytic effects are related to the position of nickel.”
If the stability of the tested catalysts was observed for at least 8 hours and for some samples - for at least 20 hours continuous operation in the bicyclohexyl dehydrogenation reaction under ideal conditions was performed at least 3 times on each of the catalysts studied. And text gives average results. Please, I suggest that these results on the stability of the catalysts with error bars (standard deviations) for the bar graphs be added to the manuscript.
Response: The resulting reaction parameters for mono-, bi- and trimetallic catalysts are shown in Fig. 3. They remained unchanged for at least 8 h of the operation in the dehydrogenation process. The error in measuring the conversion did not exceed 3 rel. %. The necessary additions have been made in the text of the article.
If the stability of the tested catalysts was observed, the recharacterization study (XPS, XRD, EDX among other techniques) of these reused materials is highly recommended to assess whether there were changes in composition, structure and texture, compared to the materials before the reaction tests.
Response: Since the optimal composition of the catalyst is in the stage of continuous refinement, studies of changes in the composition, structure and texture compared to their initial state before the reaction tests did not have a systemic nature. The results obtained are incomplete and, unfortunately, cannot be presented in the article. In this work, we focused on the problem of studying the effect of Ni and Cr in trimetallic Pt—Ni—Cr catalytic systems deposited on the carbon carrier Sibunite (C) on the activity of platinum in the bicyclohexyl dehydrogenation reaction. However, such studies will undoubtedly be carried out for elaboration of the optimal composition of the catalyst.
Line 267 the unit of the specific surface area cm2 /g or m2 /g
Line 208 check if it is XP spectra or XPS.
Please mention throughout section 2. Results and Discussion the Supplementary Information (Fig. S1, S2 and S3; Table S1, S2 and S3) of the manuscript.
Response: The typos have been corrected. The tables and figures from SI have been referred to.